# Overwater Image Dehazing via Cycle-Consistent Generative Adversarial Network

**Shunyuan Zheng** , **Jiamin Sun, Qinglin Liu, Yuankai Qi and Jianen Yan** *

School of Computer Science and Technology, Harbin Institute of Technology, Weihai 264209, China; sawyer0503@hit.edu.cn (S.Z.); sunjiamin17@gmail.com (J.S.); qlliu@hit.edu.cn (Q.L.); yk.qi@hit.edu.cn (Y.Q.)
* Correspondence: yanjianen@hit.edu.cn

**Abstract:** In contrast to images taken on land scenes, images taken over water are more prone to degradation due to the influence of the haze. However, existing image dehazing methods are mainly developed for land-scene images and perform poorly when applied to overwater images. To address this problem, we collect the first overwater image dehazing dataset and propose a Generative Adversarial Network (GAN)-based method called OverWater Image Dehazing GAN (OWI-DehazeGAN). Due to the difficulties of collecting paired hazy and clean images, the dataset contains unpaired hazy and clean images taken over water. The proposed OWI-DehazeGAN is composed of an encoder–decoder framework, supervised by a forward-backward translation consistency loss for self-supervision and a perceptual loss for content preservation. In addition to qualitative evaluation, we design an image quality assessment neural network to rank the dehazed images. Experimental results on both real and synthetic test data demonstrate that the proposed method performs superiorly against several state-of-the-art land dehazing methods. Compared with the state-of-the-art, our method gains a significant improvement by 1.94% for SSIM, 7.13% for PSNR and 4.00% for CIEDE2000 on the synthetic test dataset.

**Keywords:** image dehazing; overwater image; generative adversarial network

## 1. Introduction

Images of overwater scenes play an important role in human image galleries. However, these images are prone to degradation due to thick mist that are often appearing over lakes, rivers, and seas. Although numerous image dehazing methods have been developed [1–5], our experiments show that these methods perform far from satisfying since they are originally designed for land scene images, of which the data distribution differs significantly.

Hazy images are usually modeled as $I(x) = J(x)t(x) + A(1 - t(x))$, where $I(x)$ and $J(x)$ are the observed hazy image and the scene, respectively [6,7]. The symbol $x$ denotes a pixel index, and $A$ is the global atmospheric light. $t(\cdot)$ denotes the transmission map, which describes the portion of light that is not scattered and reaches the camera sensors. When the haze is homogeneous, $t(\cdot)$ can be defined as: $t(x) = e^{-\beta d(x)}$, where $\beta$ is the scattering coefficient and $d(x)$ is the distance between objects and the camera.

Existing methods fall into two categories according to the type of features they used: methods based on hand-crafted features [1,2,8–12] or methods based on convolutional neural network (CNN) features [3–5,13–16]. The former generally focuses on estimating the global atmospheric light intensity $A(\cdot)$ and the transmission map $t(\cdot)$, and hence their performance are susceptible to estimation errors of $A(\cdot)$ or $t(\cdot)$. To alleviate these limitations, the latter, which is based on CNNs [17] or Generative Adversarial Networks (GANs) [18], aims to directly estimate clean images in a data-driven scheme. Although promising dehazing results have been achieved, existing CNN- or GAN- based methods

perform not well on overwater images as shown in Figure 1b,c. This is because they are designed for dehazing land scene images, which results from the difference among data distributions of land images and overwater images. Another issue is that existing image dehazing datasets [12,19–22] are dominated by land scenes, which have significant data distribution differences compared to that of overwater images.

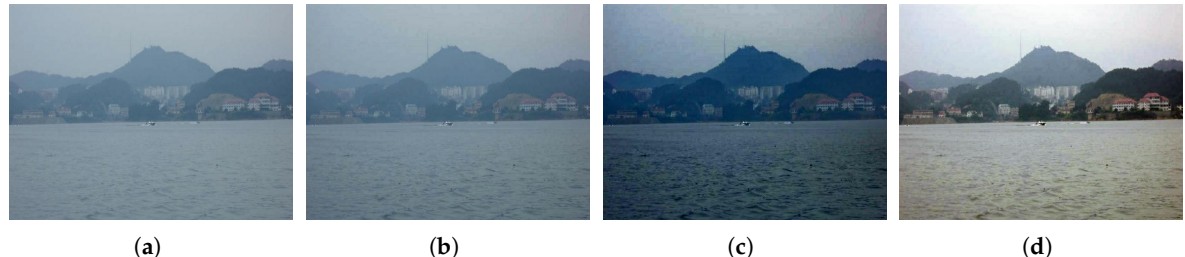

| (a) | (b) | (c) | (d) |

**Figure 1.** Overwater image dehazing example. The proposed method generates more clear images compared to state-of-the-art methods. (**a**) Hazy input, (**b**) Cai [13], (**c**) Yang [4], (**d**) ours.

In this paper, we address both the above-mentioned issues. First, we construct a new dataset, named HazyWater, especially for dehazing overwater images. Since collecting paired hazy and clear images is difficult and expensive, the HazyWater dataset is composed of unpaired hazy and clean overwater images. Although the unpaired character challenges most of the existing methods, we show that satisfying performance could be achieved by the proposed dehazing method in the experiment section. Second, we propose an OverWater Image Dehazing GAN (OWI-DehazeGAN) inspired by CycleGAN [23] to directly recover clean images. OWI-DehazeGAN employs the forward-backward translation consistency loss as self-supervision to tackle the unpairness of training data. We also introduce a perceptual loss to enhance the recovered image quality.

Our contributions are summarized as follows:

- We create the first overwater image dehazing dataset. Most of the existing image dehazing methods (including CNN-based methods) perform not well on overwater images, and we hope this dataset is able to facilitate the research in this field.
- We propose an OWI-DehazeGAN to dehaze overwater images, which is based on but performs superior to Cycle-GAN. The proposed network is able to utilize unpaired training data and preserve image details simultaneously.
- We propose an image quality assessment network to rank the dehazed images of dehazed methods, which facilitates the comparison of different dehazing methods. Extensive experimental results evaluated by this network as well as the peak signal-to-noise ratio (PSNR), structural similarity (SSIM), and CIEDE2000 metrics demonstrate the effectiveness of the proposed OWI-DehazeGAN.

## 2. Related Work

### 2.1. Methods Based on Hand-Crafted Features

Many efforts have been devoted to image dehazing in the past decades based on hand-crafted features [1,2,8–12]. Tan et al. [8] proposed a contrast maximizing approach using Markov random fields (MRF) based on the observation that clean images have higher contrast than hazy ones. In Reference [9] Tarel et al. propose a fast dehazing method by combining atmospheric veil inference, image restoration and smoothing tone mapping. Later, He et al. [10] estimated the transmission map by utilizing dark-channel prior (DCP). Meng et al. [11] explores the inherent boundary constraint on the transmission function. In order to recover depth information, Zhu et al. [1] proposed a color attenuation prior (CAP) by creating a linear model on local priors. Different from previous methods that use various patch-based priors, Berman et al. [2] presented a new image dehazing algorithm based on non-local prior so that a haze-free image is able to be well approximated by a few distinct colors.

While the afore-mentioned methods have achieved promising results, they perform far from satisfying when applied to overwater images. For example, MRF [8] tends to produce over-saturated images. The enhanced images of FVR [9] often contain distorted colors and severe halos. DCP [10] does not work well when it comes to the sky regions, as the scene objects are similar to the atmospheric light. This is mainly because the major region of overwater images are water and sky, which have a different data distribution from that of land scene images.

*2.2. Methods Based on CNN Features*

Deep convolutional neural networks have shown promising success in various computer vision tasks [24,25]. Many CNN-based image dehazing methods have been proposed. Cai et al. [13] propose an end-to-end DehazeNet with non-linear regression layers to estimate medium transmission. Instead of estimating the transmission map or the atmospheric light firstly, AOD-Net [16] predicts the haze-free images directly using a light-weight CNN. Proximal Dehaze-Net [4] combines the advantages of traditional prior-based dehazing methods and deep learning methods by incorporating haze-related prior learning into the deep network.

Since Goodfellow [18] proposed the GAN method in 2014, there have been many effective variants tailored to different computer vision tasks [23,26,27]. Motivated by the success of GANs in style transfer [28], super-resolution [29], text to image [30], and image inpainting [31], many GAN-based methods have been proposed for image dehazing. In Reference [5], a Densely Connected Pyramid Dehazing Network (DCPDN) is proposed to jointly learn the transmission map, atmospheric light and dehazing result all together. Yang et al. [32] propose to loose the paired training constraint by introducing a disentanglement and reconstruction mechanism. Li et al. [14] designed a solution based on a cGAN network [26] to directly estimate the clean image. Ren et al. [3] adopt an ensemble strategy to take advantage of the information in white balance, contrast-enhancing, and gamma correction images. Overall, these methods are trained on paired data, which is unsuitable for the proposed overwater image dehazing task, where only unpaired training data is available.

*2.3. Image Dehazing Dataset*

Image dehazing tasks profit from the continuous efforts for large-scale data. Several datasets [12,19–22] have been introduced for image dehazing. Fattal et al. [12] provide 12 high-quality synthetic images for outdoor image dehazing. For evaluating the performance of automatic driving systems in various hazy conditions, FRIDA [33] introduces a dataset that includes synthetic images. MSCNN [15] and AOD-Net [16] utilize the indoor NYU2 Depth Database [34] and the Middlebury Stereo database [35] to synthesize hazy images using the known depth information. O-HAZE [20] is an outdoor scenes dataset, which is composed of pairs of real hazy and corresponding clean data. I-HAZE [20] is a dataset that contains 35 image pairs of hazy and corresponding ground-truth indoor images. Foggy Cityscapes dataset [21] is a synthetic version of the Cityscapes dataset using incomplete depth information. Li [22] launched a new large-scale benchmark which is made up of synthetic and real-world hazy images, called Realistic Single Image Dehazing (RESIDE). However, most datasets are synthetic and not tailored to the problem of overwater image dehazing. Different from the above datasets, we collect a dataset that contains real data especially used for dehazing overwater images.

## 3. Proposed Method

In this section, we first introduce the overall structure of the proposed OWI-DehazeGAN, and then detail each of its components. At the end of this section, we present the proposed image quality ranking network, which is later used to rank the outputs of different dehazing methods in the experiment section.

Figure 2 shows the main architecture of the proposed OWI-DehazeGAN. Unlike traditional GANs, OWI-DehazeGAN consists of two generators ($G$ and $F$) and two discriminators ($D_x$ and $D_y$) in order to be trainable with unpaired training data. Specifically, generator $G$ predicts clean images $Y$ from

hazy images $X$, and $F$ vice versa. $D_x$ and $D_y$ distinguish hazy images and clean images, respectively. Below we provide more details about each component.

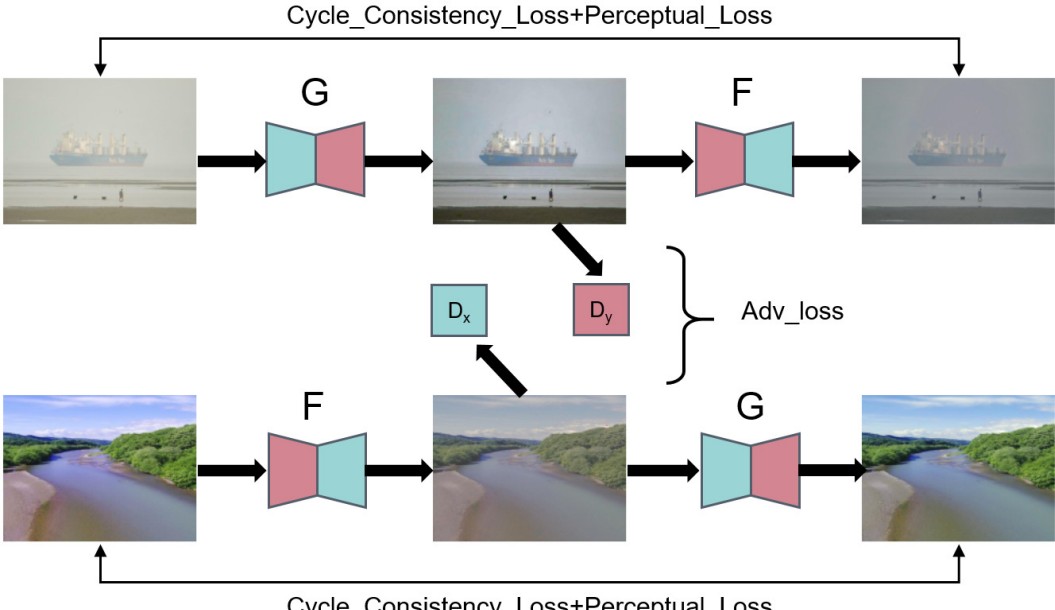

**Figure 2.** The main architecture of the proposed OverWater Image (OWI)-Dehazing network. $G$ and $F$ denote generators, where $G : X \rightarrow Y$ generates clean images from hazy images and $F : Y \rightarrow X$ vice versa. $D_x$ and $D_y$ denote discriminators. Adversarial loss, cycle consistency loss and perceptual loss are employed to train the network.

*3.1. Generator*

We adopt the same structure for the two generators $G$ and $F$. Both generators are divided into three parts: encoding, transformation, and decoding. The architecture and details of the generator are shown Figure 3a and Table 1, respectively.

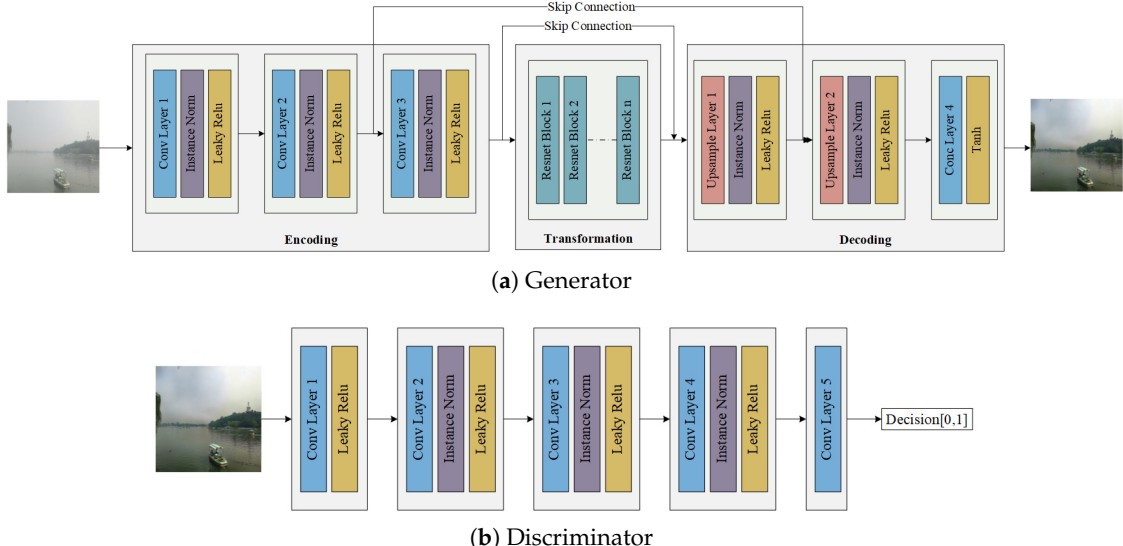

**Figure 3.** Architecture of our generator and discriminator. The generator consists of encoding, transformation, and decoding three parts.

**Table 1.** Main parameters of our generator. '#In' denotes the number of input channel, and '#Out' represents the number of output channel. 'Upsample' denotes the resize convolution.

| Layer Name | #In | #Out | Kernel | Strides |
|---|---|---|---|---|
| Conv 1 | 3 | 32 | $7 \times 7$ | 1 |
| Conv 2 | 32 | 64 | $3 \times 3$ | 2 |
| Conv 3 | 64 | 128 | $3 \times 3$ | 2 |
| Res Block 1 | 128 | 138 | $3 \times 3$ | 1 |
| ... | ... | ... | ... | ... |
| Res Block 9 | 128 | 128 | $3 \times 3$ | 1 |
| Upsample 1 | 256 | 64 | $3 \times 3$ | 2 |
| Upsample 2 | 128 | 32 | $3 \times 3$ | 2 |
| Conv 4 | 32 | 3 | $7 \times 7$ | 1 |
| Tanh | - | - | - | - |

**Encoding**: The encoding module extracts image features by three convolution layers, which serve as down-sampling layers to decrease the resolution of the original input. Each convolution layer is followed by an instance normalization [36] and Leaky Relu [37]. Since image dehazing can be treated as a domain adaptation problem that views each image as a domain, instance normalization is more suitable for image dehazing than batch normalization [38]. Leaky Relu is an improved activation method for Relu [39] which has all the benefits of Relu and solves the dead Relu problem.

**Transformation**: The transformation module translates information from one domain to another via nine ResNet blocks [40]. ResNet block in our network contains two $3 \times 3$ convolution layers with the same number of filters. Due to the results of image dehazing, the need to retain the characteristics of the original image, such as the shape and color, the ResNet block is well-suited to accomplish these transformations.

**Decoding**: The decoding module includes the up-sampling operations and nonlinear mappings. There are several choices for upsampling, such as deconvolution [41], sub-pixel convolution [42] and resize convolution [43]. In order to reduce checkerboard artifacts [43] caused by deconvolution or sub-pixel convolution, we use the resize convolution for decoding. It increases the resolution of the feature map using nearest-neighbor interpolation followed by convolution. Inspired by the success of U-Net [44], we introduce two symmetric skip connections to deliver information between encoding and decoding modules. Finally, images are recovered through convolution and tanh activation.

*3.2. Discriminator*

We use two discriminators $D_x$ and $D_y$ to distinguish the input hazy images and clean images, respectively. The discriminator is implemented in a fully convolutional fashion. We use four convolution blocks in discriminator. The first block consists of convolution and Leaky Relu, the last block only contains convolution and the remaining blocks are composed of convolution, instance normalization and Leaky Relu. In order to maintain the stability of GAN training process, we do not utilize the sigmoid activation in the last block. The architecture and details of the discriminator are shown in Figure 3b and Table 2.

**Table 2.** Main parameters of our discriminator. '#In' denotes the number of input channel, and '#Out' represents the number of output channel.

| Layer Name | #In | #Out | Kernel | Strides |
|---|---|---|---|---|
| Conv 1 | 3 | 32 | $4 \times 4$ | 2 |
| Conv 2 | 32 | 64 | $4 \times 4$ | 2 |
| Conv 3 | 64 | 128 | $4 \times 4$ | 2 |
| Conv 4 | 128 | 256 | $4 \times 4$ | 2 |
| Conv 5 | 256 | 1 | $4 \times 4$ | 1 |

### 3.3. Loss Function

We utilize three kinds of losses to train the proposed network, including Adversarial Loss, Cycle Consistency Loss, and Perceptual Loss. The Adversarial Loss and the Cycle Consistency Loss enable the proposed network trainable with unpaired data, and the Perceptual Loss preserves image details.

### 3.3.1. Adversarial Loss

As done in CycleGAN, we use adversarial loss and cycle consistency loss for unpaired training data. $x \in X$, $y \in Y$ are hazy image and clean image unpaired data, respectively. For the generator $G$ and discriminator $D_y$, the adversarial loss is formulated as:

$$L_{GAN}(G, D_y, x, y) = log(D_y(y)) + log(1 - D_y(G(x))).$$ (1)

Correspondingly, the constraint on generator $F$ and its discriminator $D_x$ is

$$L_{GAN}(F, D_x, x, y) = log(D_x(x)) + log(1 - D_x(F(y))).$$ (2)

However, the above losses are prone to unstable training and generating low quality images. To avoid the vanishing gradients problem and achieve high quality images, we use a least squares loss [27] instead of the negative log likelihood objective [18]. Therefore, Equations (1) and (2) are modified as:

$$L_{adv}(G, D_y, x, y) = \frac{1}{2} * [(D_y(G(x)) - 1)^2 + (D_y(y) - 1)^2 + D_y(G(x))^2]$$ (3)

$$L_{adv}(F, D_x, x, y) = \frac{1}{2} * [(D_x(F(y)) - 1)^2 + (D_x(x) - 1)^2 + D_x(F(y))^2].$$ (4)

The final adversarial loss is denoted as:

$$L_{adv}(G, F, D_x, D_y) = L_{adv}(G, D_y, x, y) + L_{adv}(F, D_x, x, y).$$ (5)

### 3.3.2. Cycle Consistency Loss

CycleGAN introduces a cycle consistency loss to solve the problem that an adversarial loss alone cannot guarantee that the output distribution matches the target distribution. For each image $x$, $F(G(x))$ is able to bring $G(x)$ back to the original image. Similarly, $G(F(y))$ is able to bring $F(y)$ back to the original image $y$. $F(G(x))$ is the cyclic image of input $x$, and $G(F(y))$ is the cyclic image of the original image $y$. To train generators $G$ and $F$ at the same time, the consistency loss includes two constraints: $F(G(x)) \approx x$, $G(F(y)) \approx y$. Cycle consistency loss is defined to calculate L1-norm between the input and cyclic image for unpaired image dehazing:

$$L_{cyc}(G, F) = ||F(G(x)) - x||_1 + ||G(F(y)) - y||_1.$$ (6)

### 3.3.3. Perceptual Loss

Inspired by the success of perceptual loss in style transfer [45], we introduce perceptual loss to restrict the reconstruction of image details. Instead of measuring per-pixel difference between the images, perceptual loss is concerned with the distinction between feature maps, which comprises various aspects of content and perceptual quality. The perceptual loss is defined as:

$$L_{per}(G, F) = ||\theta(x) - \theta(F(G(x)))||_2^2 + ||\theta(y) - \theta(G(F(y)))||_2^2.$$ (7)

Here, $\theta$ represents the feature maps generated from the relu4_2 on pertained VGG-16 [46] network.

### 3.3.4. Objective Function

Our final loss is defined as the weighted sum of previous losses:

$$L(G, F, D_x, D_y) = L_{adv}(G, F, D_x, D_y) + \lambda_1 L_{cyc}(G, F) + \lambda_2 L_{per}(G, F), \tag{8}$$

where coefficients $\lambda_1$ and $\lambda_2$ represent the weights of cycle consistency loss and perceptual loss, respectively. We found that giving an over-weight to perceptual loss causes the instability of training process, which could make the GAN model non-convergent, thus the weight of perceptual loss is much less than the weight of cyclic consistency loss. In training process, we minimize the generators $G$, $F$ and maximize the discriminators $D_x$, $D_y$.

The final objective function is:

$$< G^*, F^* >= \arg \min_{G,F} \max_{D_x, D_y} L(G, F, D_x, D_y). \tag{9}$$

We summarize the training procedure in Algorithm 1.

---

**Algorithm 1** OWI-DehazeGAN training procedure pseudocode.

---

**Input:** The hazy training dataset $X$; The clean training dataset $Y$; Training epoch number *epoch*.
**Output:** The well-trained generators $G$ and $F$; The well-trained discriminators $D_x$ and $D_y$.

  1: **for** $epoch = 1; epoch \leq epoch_{max}; epoch + +$ **do**
  2: 　　Draw mini-batches of samples $\{x^{(1)}, \cdots, x^{(m)}\}$ from $X$;
  3: 　　Draw mini-batches of samples $\{y^{(1)}, \cdots, y^{(m)}\}$ from $Y$;
  4: 　　**for** each mini-batch **do**
  5: 　　　　Compute the discriminator loss on real images;
  6: 　　　　Compute the discriminator loss on fake images;
  7: 　　　　Update the discriminators $D_x$ and $D_y$;
  8: 　　**end for**
  9: 　　**for** each mini-batch **do**
 10: 　　　　Compute $X \rightarrow G(x) \rightarrow F(G(x))$ and $Y \rightarrow F(Y) \rightarrow G(F(Y))$ generator loss in Equation (6);
 11: 　　　　Compute the perceptual loss in Equation (7);
 12: 　　　　Update the generators $G$ and $F$;
 13: 　　**end for**
 14: **end for**

---

### 3.4. Dehazed Image Quality Assessment

In order to verify the effectiveness of the proposed OWI-DehazeGAN, we design a dehazed image quality assessment model based on natural image statistics and the VGG network. Natural image refers to the image directly obtained from the natural scene by using optical photographic instruments such as cameras. Natural images are different from the distorted image. Natural images are directly captured from natural scenes, so they have some natural properties. By making statistics on these properties, natural scene statistics (NSS [47]) of images can be obtained. Due to the differences between natural images and distorted images in NSS, NSS has been widely used in image quality assessment, especially no-reference image quality assessment.

In this paper, the NSS we use is mean substracted contrast normalization (MSCN [48]) coefficients, which is used to normalize a hazy image. After normalization pixel intensities of haze-free by MSCN coefficient follow a Gaussian Distribution while pixel intensities of hazy images do not. The deviation of the distribution from an ideal bell curve is therefore a measure of the amount of distortion in the

image. To calculate the MSCN Coefficients, the image intensity $I(i,j)$ at pixel $(i,j)$ is transformed to the luminance $\widehat{I}(i,j)$. $\widehat{I}(i,j)$ is defined as:

$$\widehat{I}(i,j) = \frac{I(i,j) - \mu(i,j)}{\sigma(i,j) + C},$$

(10)

where $\mu(i,j)$ and $\sigma(i,j)$ represent the local mean field and local variance field obtained by calculating the image using a gaussian window with a specific size. Local Mean Field $\mu$ is the Gaussian Blur of the input image. Local Variance Field $\sigma$ is the Gaussian Blur of the square of the difference of original image and $\mu$. $C$ is a constant, in case the denominator is zero. The calculation of MSCN coefficient is shown in Figure 4. When dehazed image is normalized by MSCN coefficient, only uniform appearance and edge information are retained. Human eyes are very sensitive to edge information, so the normalized image is consistent with human vision.

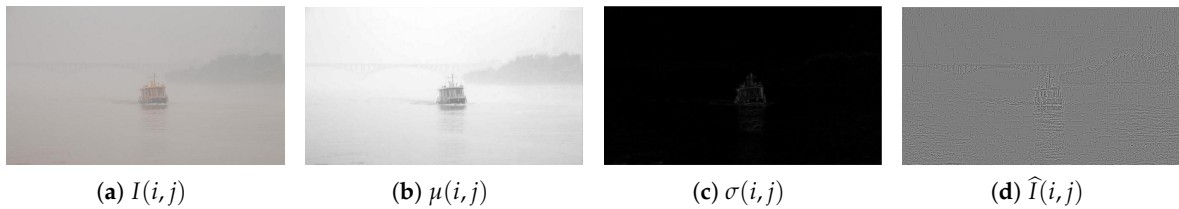

(**a**) $I(i,j)$      (**b**) $\mu(i,j)$      (**c**) $\sigma(i,j)$      (**d**) $\widehat{I}(i,j)$

**Figure 4.** Examples of mean substracted contrast normalization (MSCN) coefficient.

The proposed image quality assessment (IQA) model for dehazed images is divided into three parts: luminance normalization, feature extraction and regression of evaluation score. The dehazed images are firstly normalized by the MSCN coefficient, which provides a good normalization of image luminance and does not have a strong dependence on the intensity of texture. Then, taking the normalized image into the convolution layers of VGG-16 to extract features, and finally predicting an image quality score between 0 and 9 through two fully connection layers. The units of two fully connection layers are 512 and 1, respectively. The architecture of the IQA model is shown in Figure 5.

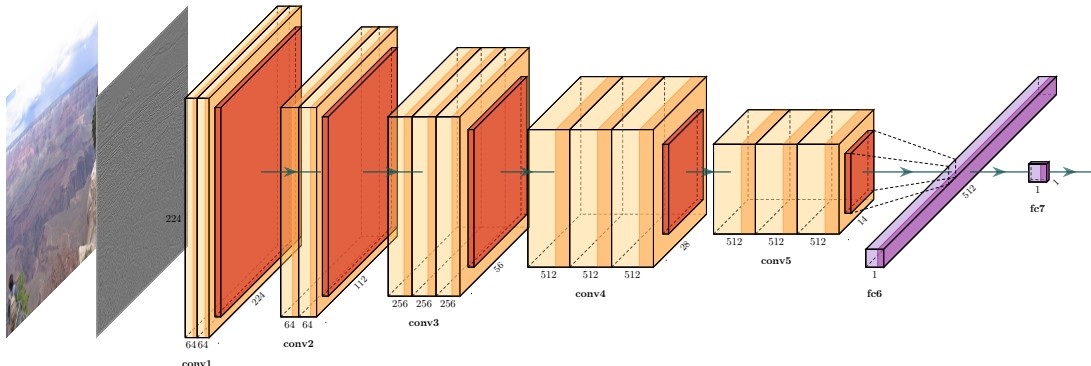

**Figure 5.** Architecture of the proposed image quality assessment (IQA) model for dehazed images.

The loss function of this IQA model is MAE. The loss is defined as:

$$loss_{IQA} = \frac{1}{N} \sum_{i=1}^{n} |y_i - y_i^*|,$$

(11)

where $N$ represents the number of images in the training set, $y_i$ and $y_i^*$ denote target data and output data, respectively. The optimization goal of the IQA model in the training phase is to minimize the average absolute error loss. Learning the mapping between dehazed images and corresponding Mean

Opinion Scores (MOS [49]) is achieved by minimizing the loss between the predicted score $y_i^*$ and the corresponding ground truth $y_i$.

## 4. Experiment

In this section, we first present our HazyWater dataset. Then we quantitatively evaluate the OWI-DehazeGAN on a synthetic dataset and real-world hazy data, with comparisons to several state-of-the-art methods. Experimental details are also explained in this section.

### 4.1. Dataset

We collect a real unpaired image dataset called HazyWater Dataset for the image dehazing in overwater scenes. All the data were gathered from Google. The training set consists of 4531 unpaired images, which are 2090 hazy images and 2441 clean images. These training images were all resized to $640 \times 480$. Figure 6 illustrates some examples of our dataset. There were three main differences between the proposed dataset and the existing datasets: (1) the HazyWater dataset is a large-scale natural real dataset with hazy images and unpaired haze-free images, while the previous datasets are only composed of synthetic data; (2) the HazyWater dataset is tailored to the task of overwater image dehazing, rather than focusing on indoor or outdoor scenes; (3) the proposed dataset is much more challenging because the regions of sky and water surface make up a large part of the image.

In order to evaluate different image dehazing methods in overwater scene subjectively, we intorduced a natural overwater testing, which contains 127 challenging hazy images collected from overwater scenes. To quantitatively compare different image-dehazing methods, we selected 90 (30 images $\times$ 3 medium extinction coefficients $\beta$) overwater hazy images with corresponding ground-truth from the RESIDE OTS dataset [22]. RESIDE OTS dataset is a large scale synthetic dataset in outdoor scene with a handful of overwater images. We apply SSIM [50], PSNR [51] and CIEDE2000 [52] to the dehazed results on synthetic images. Based on the HazyWater Dataset, we compared our proposed method against several state-of-the-art dehazing methods in real and synthetic data, including: DCP [10], FVR [9], BCCR [11], CAP [1], DehazeNet [13], MSCNN [15], AOD-Net [16], dehaze-cGAN [14], Proximal Dehaze-Net [4], CycleGAN [23].

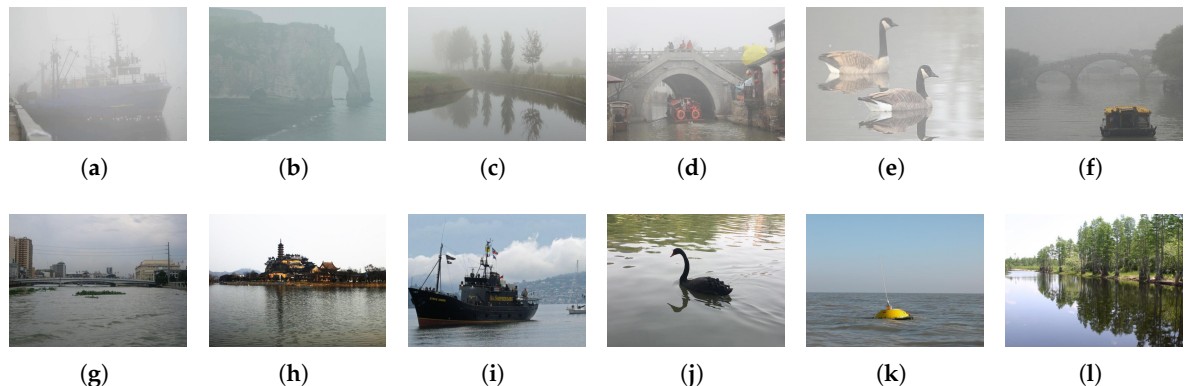

**Figure 6.** Examples of HazyWater dataset (best viewed in color). (**a**–**f**) Hazy images. (**g**–**l**) Clean imags.

### 4.2. Experimental Settings

The input images of generators and discriminators were set to $256 \times 256$ during training. We used an Adam [53] solver to optimize gradient with a learning rate of $2 \times 10^{-4}$. The batch size was 1. The weight of cyclic consistency loss $\lambda_1$ and perceptual loss $\lambda_2$ were 10 and 0.0001, respectively. The coefficient $\alpha$ of Leaky Relu was 0.2. The update proportion was 1 for generators $G$, $F$ and discriminators $D_x$, $D_y$. The proposed OWI-DehazeGAN was implemented in TensorFlow with a Nvidia GTX 1080 Ti GPU.

## 4.3. Qualitative Results on Real Images

Figures 7 and 8 show several dehazing results of the proposed algorithm against the state-of-the-art methods. DCP [10] tends to overestimate the thickness of the haze and produce dark results (Figures 7b and 8b). The dehazed images by FVR [9] and BCCR [11] have significant color distortions and miss most details as shown in Figures 7c,d and 8c,d. The best performer in the hand-crafted prior based methods was CAP [1], which generally reconstructs details of haze-free images. The deep learning-based approach achieved comparable results, such as DehazeNet [13], MSCNN [15] and dehaze-cGAN [14]. However, these results indicate that existing methods cannot handle overwater hazy images very well. For example, the dehazed results (Figures 7f,g and 8f,g) by MSCNN and DehazeNet have a similar problem that tends to magnify the phenomenon of color cast and have some remaining haze. The illumination of the Proximal Dehaze-Net [4] and AOD-Net [16] results is dark, as shown in Figures 7h,j and 8h,j, which are not consistent with human visual perception. From Figure 7k, CycleGAN generates a lot of pseudo-colors in heavy fog conditions, which is quite different from the original colors. Meanwhile, the results of CycleGAN generate extensive checkerboard artifacts in the sky regions. In contrast, the dehazed results by our method shown in Figures 7l and 8l are visually pleasing in heavy hazy or light mist condition.

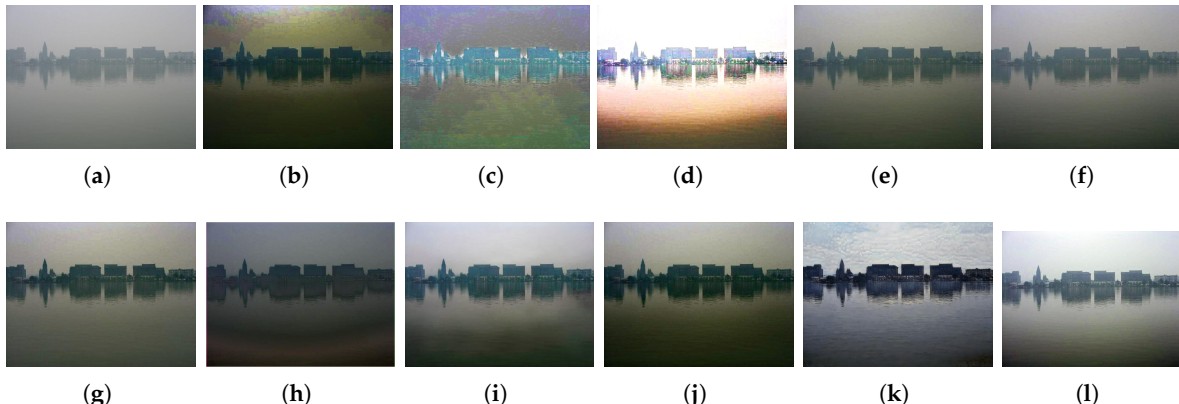

|    |    |    |    |    |    |
|----|----|----|----|----|----|
| (a) | (b) | (c) | (d) | (e) | (f) |
| (g) | (h) | (i) | (j) | (k) | (l) |

**Figure 7.** Real light hazy images and corresponding dehazing results from several state-of-the-art methods (best viewed in color). (**a**) Hazy Image. (**b**) DCP. (**c**) FVR. (**d**) BCCR. (**e**) CAP. (**f**) DehazeNet. (**g**) MSCNN. (**h**) AOD-Net. (**i**) dehaze-cGAN (**j**) Proximal. (**k**) CycleGAN. (**l**) Ours.

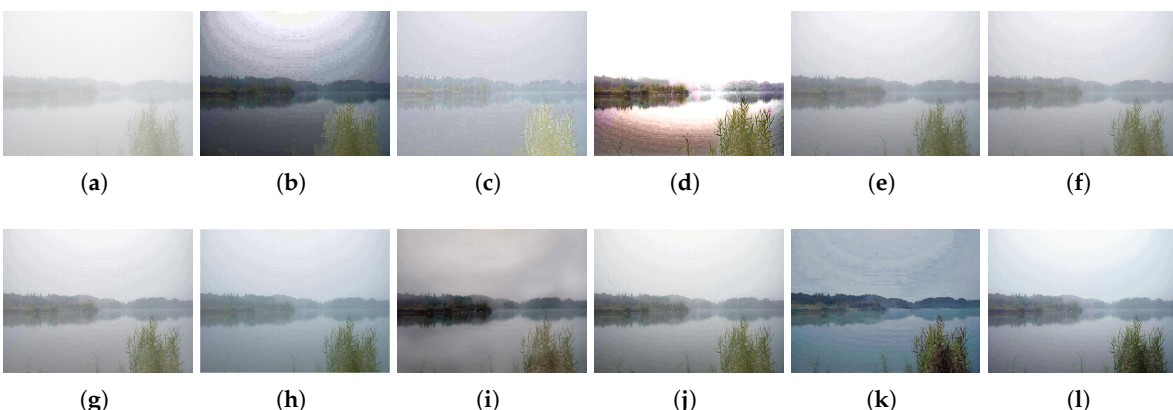

|    |    |    |    |    |    |
|----|----|----|----|----|----|
| (a) | (b) | (c) | (d) | (e) | (f) |
| (g) | (h) | (i) | (j) | (k) | (l) |

**Figure 8.** Real heavy hazy images and corresponding dehazing results from several state-of-the-art methods (best viewed in color). (**a**) Hazy Image. (**b**) Dark-channel prior (DCP). (**c**) FVR. (**d**) BCCR. (**e**) Color attenuation prior (CAP). (**f**) DehazeNet. (**g**) MSCNN. (**h**) AOD-Net. (**i**) dehaze-cGAN (**j**) Proximal. (**k**) CycleGAN. (**l**) Ours.

### 4.4. Qualitative and Quantitative Results on Synthetic Images

We further conducted some experiments based on synthetic hazy images. Although the proposed method is trained on real unpaired data, we note that it can be applied for synthetic images as well. Figure 9 shows some dehazed images generated by various methods. Figure 9a shows the groundtruth as reference. As shown in Figure 9b–d, the results of DCP [10], FVR [9] and BCCR [11] have some color or detail distortion. The dehazed results by CAP [1] (Figure 9e), DehazeNet [13] (Figure 9f), MSCNN [15] (Figure 9g), AOD-Net [16] (Figure 9h), dehaze-cGAN [14] (Figure 9i) and Proximal Dehaze-Net [4] (Figure 9j) are closer to groundtruth Figure 9a than the results based on priors. However, there was still some remaining haze as shown in Figure 9e–h. The result generated by CycleGAN in Figure 9k shows that there exists serious color cast and losses of some color information. The dehazed result generated by our approach in Figure 9l, by contrast, is visually close to the groundtruth image.

An advantage of testing on synthetic data is able to objectively evaluate experimental results via SSIM, PSNR and CIEDE2000. SSIM, PSNR and CIEDE2000 provide a pixel-wise measure between clean images and dehazed images. A higher SSIM score indicates that the generated results are more consistent with human perception. PSNR forecasts the effectiveness of image dehazing, and CIEDE2000 presents that smaller scores indicate better color preservation. In Figure 9, the SSIM and PSNR values also indicate that our method surpasses other methods. From Table 3, our method get higher PSNR, higher SSIM and lowerCIEDE2000 on the synthetic dataset. Compared with the state-of-the-art, our method gained a significant improvement by 1.94% for SSIM, 7.13% for PSNR and 4.00% for CIEDE2000, and significantly better than CycleGAN. In addition, the generator F of our proposed method can be used to generate paired image dehazing data, as shown in Figure 10. In general, the dehazed results by the proposed algorithm had higher visual quality and fewer color distortions.

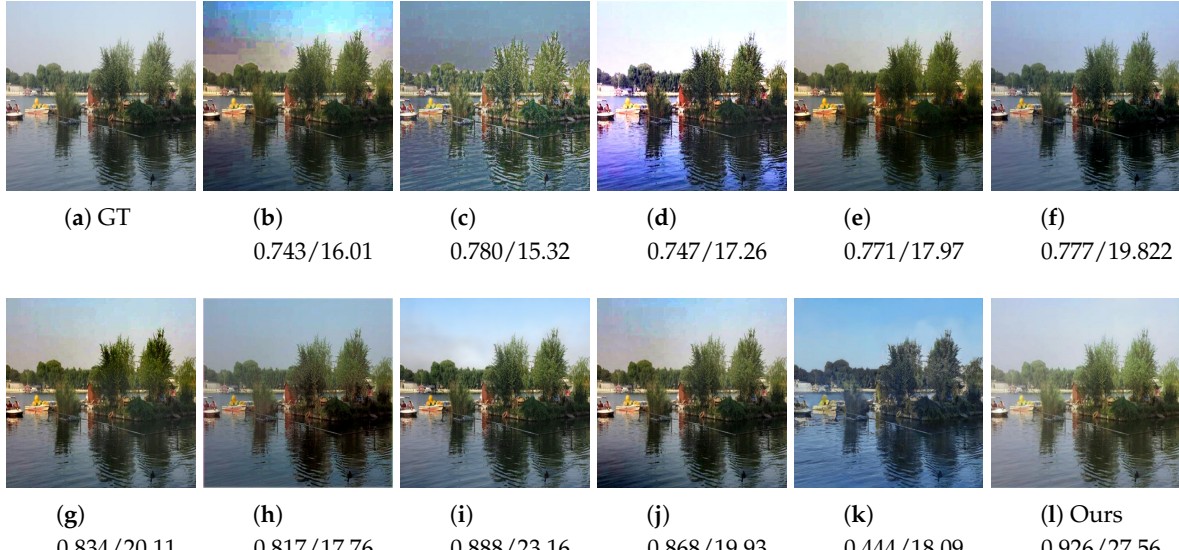

**Figure 9.** Comparison in terms of SSIM/PSNR for different image dehazing methods. (**a**) Ground Truth. (**b**) DCP. (**c**) FVR. (**d**) BCCR. (**e**) CAP. (**f**) DehazeNet. (**g**) MSCNN. (**h**) AOD-Net. (**i**) dehaze-cGAN. (**j**) Proximal. (**k**) CycleGAN. (**l**) Ours.

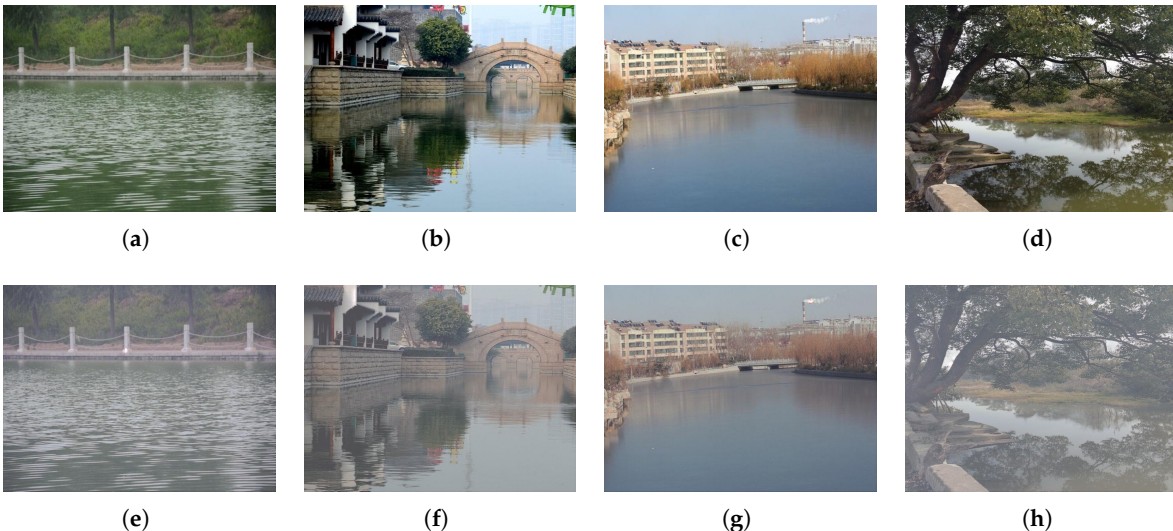

**Figure 10.** Examples of paired data generated by our network. (**a–d**) Clean images. (**e–h**) Generated hazy images corresponding to (**a–d**).

**Table 3.** Average PSNR, SSIM, and CIEDE2000 values of dehazed results on the new synthetic dataset. The best result, the second result, and the third place result are represented by red, blue, and green, respectively. The average values of PSNR , SSIM and CIEDE2000 results calculated directly between the each hazy and its corresponding clean image.

|  | FVR | DCP | BCCR | CAP | DehazeNet | MSCNN | AOD-Net | Proximal | dehaze-cGAN | CycleGAN | Ours |
|---|---|---|---|---|---|---|---|---|---|---|---|
| SSIM | 0.817 | 0.717 | 0.680 | 0.825 | 0.867 | 0.850 | 0.835 | 0.820 | 0.876 | 0.584 | 0.893 |
| PSNR | 16.56 | 14.57 | 13.92 | 19.50 | 23.27 | 20.55 | 19.33 | 17.79 | 22.77 | 18.31 | 24.93 |
| CIEDE2000 | 11.71 | 14.92 | 15.35 | 9.37 | 6.23 | 7.81 | 9.11 | 10.14 | 6.51 | 11.5 | 5.98 |

## 4.5. Dehazed Image Ranking

The proposed IQA model for dehazed images was pre-trained on TID2013 [49] and then fine-tuned using the IVC Dehazing Dataset [19]. The TID2013 includes different types of image distortion, while IVC Dehazing Dataset is designed to evaluate the quality of dehazed images. Predicted scores were used to qualitatively rank photos as shown in Figure 11. Ranking scores and the rankings are presented below each image, where '1' denotes the best visual perception and '10' for the worst. Figure 11 shows that the quality of overwater dehazed images generated by OWI-DehazeGAN is better than other methods. For a comprehensive comparison, we also report the dehazed image quality measured by four typical image quality assessment methods in Table 4. The best result is shown in red font. Table 4 shows that the proposed method achieves the best performance in terms of almost all metrics.

**Table 4.** Comparison of dehazed image quality using four image quality assessment methods. The top three results are in red, blue, and green font, respectively.

|  | FADE [48] ↓ | SSEQ [54] ↓ | BLINDS-2 [55] ↓ | NIMA [56] ↑ |
|---|---|---|---|---|
| Ours | 1.95 | 36.24 | 31.50 | 6.47 |
| CAP | 3.03 | 40.97 | 51.00 | 5.95 |
| MSCNN | 2.38 | 45.56 | 50.50 | 6.23 |
| DehazeNet | 3.11 | 42.49 | 49.50 | 6.12 |
| Proximal-Dehaze | 1.69 | 43.69 | 54.50 | 6.06 |
| DCP | 1.45 | 43.32 | 51.50 | 5.36 |
| BCCR | 1.17 | 45.76 | 48.50 | 6.08 |
| FVR | 2.67 | 41.71 | 48.00 | 4.86 |
| AOD-Net | 2.30 | 38.62 | 54.50 | 6.15 |

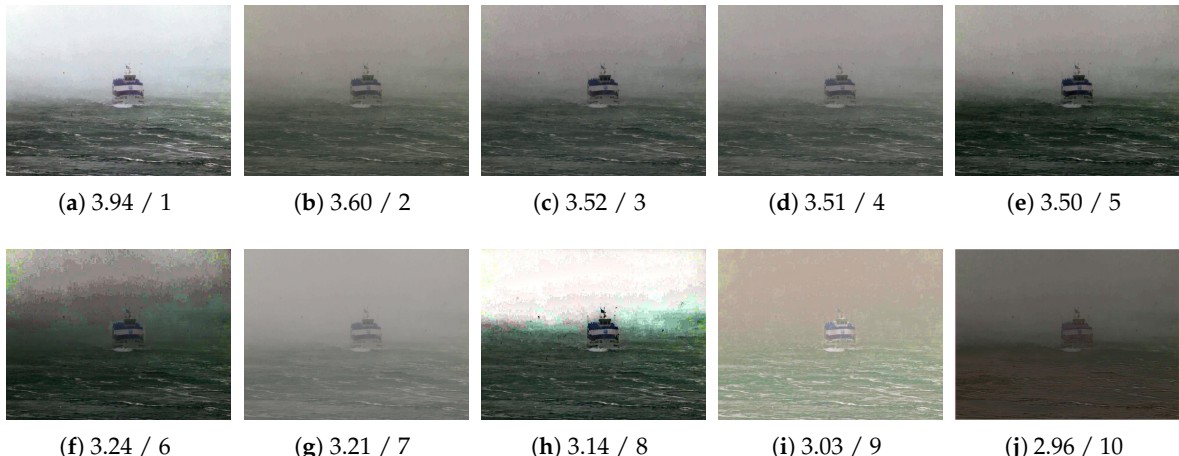

**Figure 11.** Comparison via the proposed IQA model. Ranking scores and the rankings are shown below each image. (**a**) Ours. (**b**) CAP. (**c**) MSCNN. (**d**) DehazeNet. (**e**) Proximal. (**f**) DCP. (**g**) Input. (**h**) BCCR. (**i**) FVR. (**j**) AOD-Net.

*4.6. Run Time*

We also conducted the running speed experiment to evaluate the efficiency of each method. For fairness, we ran all methods on the same platform (Intel Core i7-7800X CPU@3.50 GHz, 32 GB memory). In the experiment, 90 synthetic images were used for testing, each of which were resized to 512 × 512 pixels. DCP [10], FVR [9] and CAP [1] are run without GPU acceleration. DehazeNet [13], MSCNN [15], AOD-Net [16], dehaze-cGAN [14] and our model run with a GPU card (NVIDIA GeForce GTX 1080 Ti). Table 5 shows more implementation details and the average run time. AOD-Net [16] ran fastest thanks to its lightweight structure. Our method ran slightly slower than AOD-Net (0.07 s behind) but ran 0.34 s faster than the third best method. Overall, the proposed method performed comparably against state-of-the-art methods in terms of the running speed.

**Table 5.** Average run time (in seconds) on synthetic data. The top three results are highlighted in red, blue, and green font, respectively.

| Methods | Time (512 × 512) | Run Environment |
|---|---|---|
| DCP | 15.55 | Matlab |
| FVR | 3.87 | Matlab |
| CAP | 0.94 | Matlab |
| DehazeNet | 1.02 | Matlab |
| MSCNN | 1.97 | Matlab |
| AOD-Net | 0.23 | PyTorch |
| dehaze-cGAN | 0.64 | Torch7 |
| Ours | 0.30 | TensorFlow |

## 5. Analysis and Discussion

In this section, we further explore and analyze the effect of each component of the proposed method. Firstly, we explore the robustness of the proposed method in image color cast removal. Then, we discuss the advantage of resizing convolution comparing to other up-sampleing methods. We also analyze the effect of perceptual loss for image the dehazing task. Finally, we show the limitation of the proposed algorithm.

*5.1. Robustness to Color Cast*

In the task of image dehazing, if the global atmospheric light is estimated incorrectly, it is easy to cause image color cast. The proposed OWI-DehazeGAN is robust to color cast removal. Existing

methods, such as DehazeNet and MSCNN, will amplify this disadvantage and reduce image quality. An effective measure to solve the image color cast problem is color balance. The purpose of color balance is to make sure that the intrinsic color of the objects in the scene does not change under different illumination conditions. Our method plays an important role in color balance which can remove the color cast. As shown in Figure 12, our method performed well when the image encounters color cast. Compared to DCP, DehazeNet and MSCNN, our method effectively corrects the color cast phenomenon and produces a better visual image. Although our method cannot produce completely haze-free results with the best visual quality, while properly retaining a certain haze is beneficial to perceiving image depth.

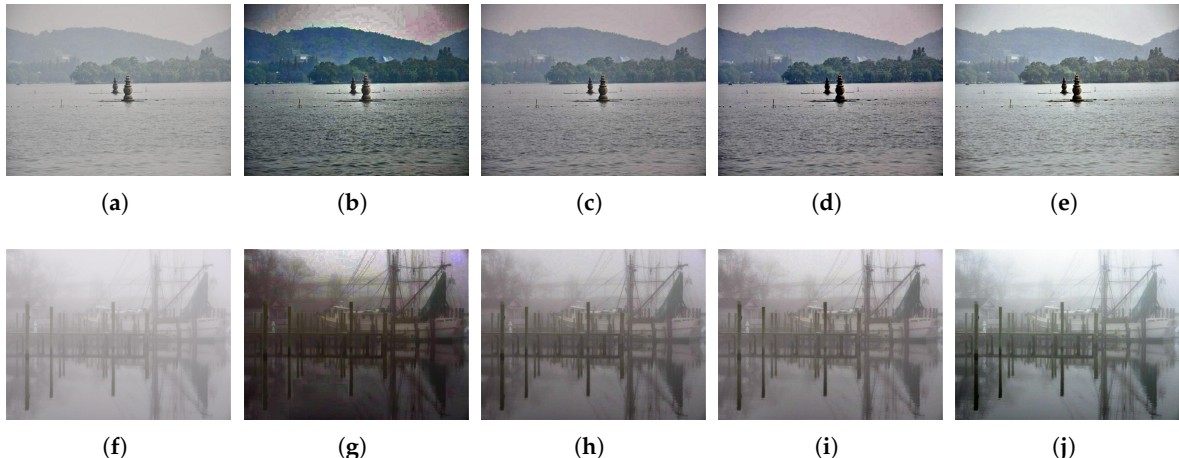

(**a**)　　　　(**b**)　　　　(**c**)　　　　(**d**)　　　　(**e**)

(**f**)　　　　(**g**)　　　　(**h**)　　　　(**i**)　　　　(**j**)

**Figure 12.** Examples of color cast. (**a**,**f**) Two hazy images. (**b**,**g**) Generated by DCP. (**c**,**h**) Generated by DehazeNet. (**d**,**i**) Generated by MSCNN. (**e**,**j**) Our results. Compared with DCP, DehazeNet and MSCNN, our method effectively corrects the color cast phenomenon and produces a visual better image.

### 5.2. Effect of Resize Convolution

In the decoding process of the generator, we use resize convolutions to increase the resolution of the feature maps, rather than deconvolution or sub-pixel convolution. To better understand how the resize convolution contributes to our proposed method, we train three end-to-end networks with different upsample mode: (i) deconvolution, (ii) sub-pixel convolution, and (iii) resize convolution.

Figure 13 shows the results of the deconvolution, sub-pixel convolution, and resize convolution as the upsampling mode in our network. In comparison, the results of resize convolution are best viewed from the perspective of the human perception and retain more detailed information. It can be seen from Figure 13b, that there are many artifacts in the regions of sky and water surface. From Figure 13f which is the zoom-in views in Figure 13b, we observe plenty checkerboard pattern of artifacts caused by deconvolution. Although the sub-pixel convolution (Figure 13c) can alleviate the "checkerboard artifacts" to some extent, the result of sub-pixel convolution is rough and unsatisfying. Comparing with the first two approaches, resize convolution recovers most scene details and maintain the original colors. Furthermore, it is obvious that the result of resize convolution is the best from the comparison of the zoom-in views. From Table 6, the introduced resize convolution gains higher PSNR, SSIM scores and a lower CIEDE2000 score than deconvolution and sub-pixel convolution, which indicate that resize convolution can generate visually perceptible images.

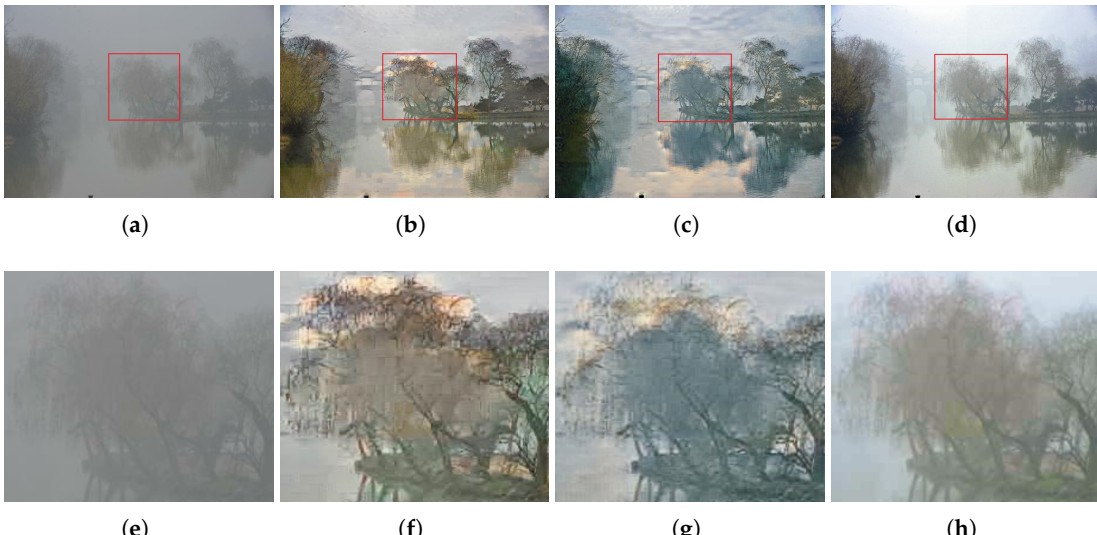

**Figure 13.** Effectiveness of the proposed network with resize convolution. (**a**,**e**) Input hazy images. (**b**,**f**) Dehazed images using deconvolution. (**c**,**g**) The dehazed images using sub-pixel convolution. (**d**,**h**) Denote dehazed results using resize convolution (ours). (**e**–**h**) The zoom-in views of (**a**–**d**), separately.

**Table 6.** Average scores in terms of PSNR, SSIM, and CIEDE2000 for deconvolution, sub-pixel convolution, and resize convolution on the synthetic test set from HazyWater Dataset.

| Average Metrics | SSIM | PSNR | CIEDE2000 |
|---|---|---|---|
| Deconvolution | 0.758 | 20.34 | 9.12 |
| Sub-pixel Convolution | 0.643 | 20.21 | 10.86 |
| Resize Convolution | **0.819** | **22.19** | **7.41** |

## 5.3. Effect of Perceptual Loss

In this section, we analyze how the introduced perceptual loss helps estimate high-quality haze-free images. To show the effectiveness of our loss function, we also train an overwater image dehazing network without perceptual loss. We show an example of result with and without perceptual loss as shown in Figure 14, the generated direction for images in the first row is $X \rightarrow G(X) \rightarrow F(G(X))$, the second row is the opposite of the first row ($Y \rightarrow F(Y) \rightarrow G(F(Y))$). We observed from Figure 14c,d,g,h that the estimated haze-free images and cyclic images lack fine details and the regions of the sky do not match with the input hazy image, which leads to the dehazed results containing halo artifacts when the perceptual loss is not used. Through the comparison of Figure 14b,d we can also find that the perceptual loss is favorable for the reconstruction of the sky regions, which is very necessary for the overwater image dehazing. Meanwhile, comparing Figure 14c,h, we can know that the perceptual loss has a little inhibition to the colour cast.

From Table 7, we observe that results generated by our network with perceptual loss gain higher PSNR, SSIM and lower CIEDE2000 score. Higher SSIM and PSNR scores suggest that the proposed method with perceptual loss has a better image dehazing effect and is consistent with human perception. Lower CIEDE2000 means less color difference between dehazed image and groundtruth. The above experiments show that the proposed loss is effective for overwater image dehazing.

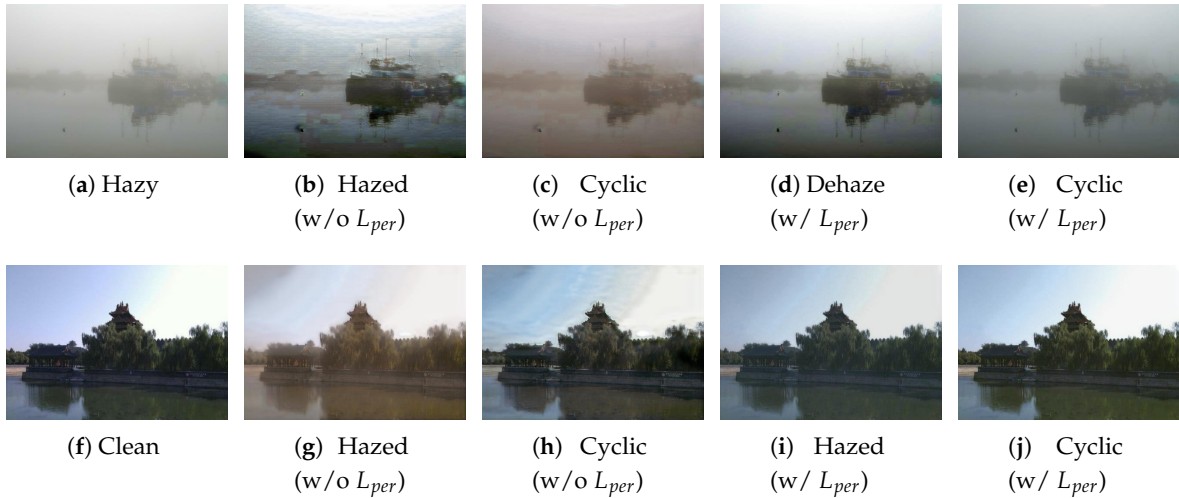

|  |  |  |  |  |
|---|---|---|---|---|
| (**a**) Hazy | (**b**) Hazed (w/o $L_{per}$) | (**c**) Cyclic (w/o $L_{per}$) | (**d**) Dehaze (w/ $L_{per}$) | (**e**) Cyclic (w/ $L_{per}$) |
| (**f**) Clean | (**g**) Hazed (w/o $L_{per}$) | (**h**) Cyclic (w/o $L_{per}$) | (**i**) Hazed (w/ $L_{per}$) | (**j**) Cyclic (w/ $L_{per}$) |

**Figure 14.** Comparison of dehazing with and without perceptual loss. (**a**–**e**) The generation direction of $X \rightarrow Y \rightarrow X$. (**f**–**j**) The direction of formation is $Y \rightarrow X \rightarrow Y$. "(w/o $L_{per}$)" denotes the network without perceptual loss, and "(w/ $L_{per}$)" denotes the network with perceptual loss.

**Table 7.** Effect of perceptual loss in terms of SSIM, PSNR, and CIEDE2000. CycleGAN loss refers to the formulation of adversarial loss and cycle consistency loss, VGG loss refers to substitute perceptual loss.

| Average Metrics | SSIM | PSNR | CIEDE2000 |
|---|---|---|---|
| CycleGAN loss | 0.819 | 22.19 | 7.41 |
| CycleGAN loss + VGG loss | **0.893** | **24.93** | **5.98** |

*5.4. Limitation*

Although the proposed algorithm is effective in most cases, a limitation of our method is that it cannot well handle degraded overwater images with very heavy haze, which are close to pure white. As shown in Figure 15, our network fail to adopt to this situation because the extremely heavy haze causes the illumination of water and sky regions to be close to atmospheric light.

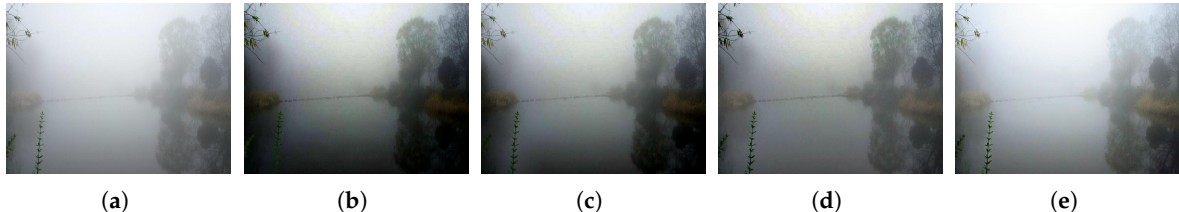

|  |  |  |  |  |
|---|---|---|---|---|
| (**a**) | (**b**) | (**c**) | (**d**) | (**e**) |

**Figure 15.** Example of failure cases. (**a**) Input image. (**b**) CAP. (**c**) DehazeNet. (**d**) MSCNN. (**e**) Ours.

## 6. Conclusions

In this paper, we formulate an overwater image dehazing task, create the first overwater image dehazing dataset, and propose the OWI-DehazeGAN to dehaze overwater images. Compared to previous CNN-based methods which require paired training data, the proposed OWI-DehazeGAN is able to be trained unpaired images. Our method directly predicts clean images from hazy input bypassing to estimate transmission maps and global atmospheric lights. We utilize the perceptual loss and resize convolution to preserve detailed textures and to alleviate the checkerboard artifacts. Extensive experimental results demonstrate that the presented method produces superior results than most state-of-the-art dehazing methods.

**Author Contributions:** S.Z., J.S. and Q.L. were responsible for the overall design of the study. S.Z. performed the experiments and drafted the manuscript. Y.Q. and J.Y. revised the manuscript. All authors read and agreed to the published version of the manuscript.

**Funding:** This work was supported in part by National Science and Technology Major Project under Grant No.2018YFB0804703, National Natural Science Foundation of China (Nos. 61902092 and 61872112), Fundamental Research Funds for the Central Universities Grant No.HIT.NSRIF.2020005, and National Key Research and Development Program of China (Nos. 2018YFC0806802 and 2018YFC0832105).

**Conflicts of Interest:** The authors declare no conflict of interest.

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
