# Peer review of "Overwater Image Dehazing via Cycle-Consistent Generative Adversarial Network"

_electronics, doi:10.3390/electronics9111877_

Round 1

Reviewer 1 Report

In the paper, the authors proposed overwater image dehazing. In general, the quality of the paper is high. I have only a few small issues:

1) In the abstract, the novelty must be more underlined. Add some words about the obtained results.

2) Some models of the training process must be added.

3) Add some pseudocode to show the implementation.

4) Experimental section should be extended:

a) add some more complex comparison with other solutions.

b) compare your proposal with other learning transfer solutions.

c)make more complex statistical analysis.

Author Response

Point 1: In the abstract, the novelty must be more underlined. Add some words about the obtained results.

Response 1: In this revision, we have improved the abstract by underlining the novelties and adding more description about the obtained results.

Point 2: Some models of the training process must be added.

Response 2: In this revision, we have added more details about the training process. 

Point 3: Add some pseudocode to show the implementation.

Response 3: We have added the pseudocode of the proposed OWI-DehazeGAN in Line 162.

Point 4: Experimental section should be extended:(a) add some more complex comparison with other solutions. (b) compare your proposal with other learning transfer solutions. (c) make more complex statistical analysis.

Response 4: In this revision, we added more analysis and comparison between the proposed method and other methods.

Reviewer 2 Report

The paper at hand introduces a technique to dehaze images with a high portion of water automatically. The authors share a new dataset of images for testing overwater image dehazing, but also compare on existing datasets. Overall, the paper is well written and ready for publication after slight modifications that can be addressed in a minor revision.

The abstract needs most editing from my perspective. Line 1 “In contrast, … “ in contrast to what? The previous sentence mentioned land and overwater scenes. Line 6: GAN is used before introduction. Lines 9ff: “Our OWI-DehazeGAN learns…” this sentence is not understandable and sounds more like a list of buzzwords than an actual description.

It is a slight disadvantage that even the new dataset, captured by the authors, is an unpaired set. In future it would be nice, although much more labor intensive, to have a paired dataset. However, this can’t be addressed in a revision and the paper is worth to be published without a paired dataset.

Line 151ff: “We utilize three kinds…” This sentence is hard do understand, the authors should revise this one.

Table 1 lists the parameters of the approach, which is great. The paper is missing a paragraph about the robustness of the proposed algorithm against changes of the parameters.

The variable \mu is used twice: Equation 8 as weight of perceptual loss, and in Equation 10 as local mean field. One of these variables should be renamed to avoid confusion.

The evaluation of the presented approach is comprehensive, the conclusion is well written and addresses the limitations of the approach. However, the evaluation is based on an automatic measure based on a VGG network. It would be good to include a small (based on a few images and some subjects) perception study, to not rely on an automated approach only. If outcome of the perception study aligns with the results of the automated measure, this would strengthen the findings clearly.

Author Response

Point 1: The abstract needs most editing from my perspective. Line 1 “In contrast, … “ in contrast to what? The previous sentence mentioned land and overwater scenes. Line 6: GAN is used before introduction. Lines 9ff: “Our OWI-DehazeGAN learns…” this sentence is not understandable and sounds more like a list of buzzwords than an actual description.

Response 1: Thanks for these suggestions. In this revision, we have revised the abstract in Line 1, Line 5, and Line 7ff.

Point 2: It is a slight disadvantage that even the new dataset, captured by the authors, is an unpaired set. In future it would be nice, although much more labor intensive, to have a paired dataset. However, this can’t be addressed in a revision and the paper is worth to be published without a paired dataset.

Response 2: Thanks for your approval, and we will seriously consider your suggestion on proposing a paired dataset in our future work.

Point 3: Line 151ff: “We utilize three kinds…” This sentence is hard do understand, the authors should revise this one.

Response 3: We are sorry for the confusion. This sentence has been revised in Line 149ff.

Point 4: Table 1 lists the parameters of the approach, which is great. The paper is missing a paragraph about the robustness of the proposed algorithm against changes of the parameters.

Response 4: The parameters listed in Table 1 are obtained by running multiple experiments and we chose the parameters that achieve the best performance. Therefore, we did not provide the analysis about the robustness of the proposed algorithm against the changes of the parameters.

Point 5: The variable \mu is used twice: Equation 8 as weight of perceptual loss, and in Equation 10 as local mean field. One of these variables should be renamed to avoid confusion.

Response 5: Thanks for pointing this out. We have renamed the coefficients in Equation 8 to avoid confusion. The variable \mu in Equation 10 remains unchanged, which still represents local mean field.

Point 6: The evaluation of the presented approach is comprehensive, the conclusion is well written and addresses the limitations of the approach. However, the evaluation is based on an automatic measure based on a VGG network. It would be good to include a small (based on a few images and some subjects) perception study, to not rely on an automated approach only. If outcome of the perception study aligns with the results of the automated measure, this would strengthen the findings clearly.

Response 6: We consider that different subjects have different adjudgments on the same images. In addition, we have provided quantitative results on the syst synthetic test set, which better demonstrate the performance of the proposed. Therefore, we did not conduct a perception study.

Reviewer 3 Report

The authors proposed an overwater image dehazing task, created the first HazyWater dataset, and proposed a OWI-DehazeGAN to dehaze overwater images. Compared with previous methods requiring paired data, the proposed method is able to be trained by unpaired images. The authors utilized the perceptual loss and resize convolution to preserve detailed textures and to alleviate the checkerboard artifacts. Experimental results demonstrate that the proposed method can produce superior results than most of state-of-the-art dehazing methods.

The paper is presented in a logical and clear manner, and it warrants publication, but it still needs some improvements.

Here are my comments:

  1. The authors mentioned "unstable training" in section 3.3.1 and section 3.3.3 Line 159-160. Can the authors explain a little bit more about "unstable training"? It would be better it the authors can provide some quantitative explanations.
  2. In section 4.1, Line 192-194, the authors mentioned "The HazyWater dataset is a large-scale natural real dataset with hazy images and unpaired hazy-free images", but in Figure 6, there is still "synthetic" in Figure caption. Is this a typo?  
  3. In Table 4, are all the scores ranked in the order of red, blue, and green? It looks the first two columns are in the order of red, green, and blue.
  4. The English writing of this manuscript needs to be significantly improved. 
  5.  

Author Response

Point 1: The authors mentioned "unstable training" in section 3.3.1 and section 3.3.3 Line 159-160. Can the authors explain a little bit more about "unstable training"? It would be better if the authors can provide some quantitative explanations.

Response 1: The “unstable training” in section 3.3.1 Line means the vanishing gradients problem of the discriminator, while in section 3.3.3 it means the non-convergent problem of the generator.

Point 2: In section 4.1, Line 192-194, the authors mentioned "The HazyWater dataset is a large-scale natural real dataset with hazy images and unpaired hazy-free images", but in Figure 6, there is still "synthetic" in Figure caption. Is this a typo?

Response 2: Thanks for figuring out this typo. We have revised the caption of Figure 6 by striking out “synthetic”, while the figure caption is now written as “Examples of the training set in HazyWater dataset (best viewed in color).”

Point 3: In Table 4, are all the scores ranked in the order of red, blue, and green? It looks the first two columns are in the order of red, green, and blue.

Response 3: Thanks for your correction, the colors of the first two columns in Table 4 have been revised, where the scores are ranked in the order of red, blue, and green.

Point 4: The English writing of this manuscript needs to be significantly improved.

Response 4: As suggested, we have revised the whole manuscript carefully and tried to avoid any grammar or syntax error. We believe that the language is now acceptable for the review process.

Round 2

Reviewer 1 Report

It can be accepted.